# Sources Causing Long-Term and Seasonal Changes in Combustion-Derived Particulate Matter in the Urban Air of Sapporo, Japan, from 1990 to 2002

**Kazuichi Hayakawa** [1,*], **Shigekatsu Sakai** [2] **and Tomoko Akutagawa** [2]

1   Institute of Nature and Environmental Technology, Kanazawa University, Kakuma-machi, Kanazawa 920-1192, Ishikawa, Japan
2   Research Institute of Energy, Environmental and Geology, Hokkaido Research Organization, Kita 19 Nishi 11, Kita-Ku, Sapporo 060-0819, Hokkaido, Japan
*   Correspondence: hayakawa@p.kanazawa-u.ac.jp

**Abstract:** Fifty-one samples were collected seasonally to estimate the amounts of total suspended particulate (TSP) in Sapporo, Japan, from 1990 to 2002. The atmospheric concentration of combustion-derived particulate ($P_c$) was calculated based on the NP method using 1-nitropyrene and pyrene. The atmospheric TSP and $P_c$ concentration ranges were between 31–121 $\mu g\ m^{-3}$ of air (Mean $\pm$ standard deviation (SD) = 58.2 $\pm$ 20.2 $\mu g\ m^{-3}$) and 31–121 $\mu g\ m^{-3}$ (Mean $\pm$ SD = 8.2 $\pm$ 6.0 $\mu g\ m^{-3}$), respectively. First-order linear regression equations suggested that the $P_c$ fraction decreased faster than TSP. The highest and lowest $P_c$ concentrations were observed in winter and summer, respectively, whereas the highest and lowest TSP concentrations were observed in spring and winter, respectively. The largest and smallest $P_c$/TSP concentration ratios were observed in winter (0.324) and summer (0.075), respectively. The seasonal fractions of high-temperature combustion-derived particulate ($P_h$) in $P_c$ ranged from 0.56 (winter) to 0.75 (summer), suggesting that the contribution of vehicle emissions to $P_c$ was always larger than those of coal and biomass combustion. The sources of long-term and seasonal change in $P_c$ were elucidated by analyzing organic source markers. Atmospheric concentrations of polycyclic aromatic hydrocarbons (PAHs), nitropolycyclic aromatic hydrocarbons (NPAHs) and hopanes showed long-term and seasonal changes similar to those of $P_c$, although biomarkers of biomass and coal combustion, such as levoglucosan, mannosan, and galactosan were not as strongly correlated. These results suggest that the change in the $P_c$ concentration was mainly affected by vehicle emissions rather than by coal and biomass combustion or secondary pollutant formation. The decrease in the $P_c$ over the study period was mainly a result of the Japanese particulate matter/NOx regulations on vehicle exhaust.

**Keywords:** total suspended particulate; combustion-derived particulate matter; NP method; polycyclic aromatic hydrocarbon; nitropolycyclic aromatic hydrocarbon; vehicle; coal/biomass combustion



## 1. Introduction

Air pollution kills millions of people every year. Among airborne pollutants, research has focused on particulate matter (PM), in particular the fine particulates (having a diameter $\leq$ 2.5 $\mu$m; $PM_{2.5}$), because of its relationship to respiratory and cardiovascular diseases [1]. $PM_{2.5}$ contains numerous carcinogens and mutagens such as polycyclic aromatic hydrocarbons (PAHs), e.g., benzo[*a*]pyrene (BaP), and nitropolycyclic aromatic hydrocarbons (NPAHs), e.g., 1-nitropyrene (1-NP) and dinitropyrenes. However, among the 16 priority PAHs determined by the United States Environmental Protection Agency (US EPA), BaP contributes only 11% to the total cancer risk. The other PAHs of the USEPA PAHs and NPAHs, respectively, contribute 72% and 17%. Moreover, 47% of the direct-acting mutagenicity of PM extracts cannot be attributed to eight of the commonly quantified NPAHs, as

the toxic equivalent is only known for a limited number of NPAHs [2,3]. On the other hand, several oxygenated PAHs, such as the hydroxy PAHs, quinoid PAHs, and hydroxy NPAHs, have demonstrated endocrine-disrupting activity, reactive oxygen species overproduction activity, and indirect-acting mutagenicity [4–6]. The biological activities of PAHs with polar functional groups are higher than those of their parent PAHs [7]. These reports indicate that PM contains many unknown hazardous chemical substances.

In addition to the materials emitted from combustion, PM contains substances generated from plants and animals. However, separating PM into combustion and noncombustion-derived fractions is challenging. The classification of PM into combustion and noncombustion-derived fractions and the knowledge of their sources can significantly facilitate the research on unknown toxic chemicals. Currently, for source analysis, several methods are used, such as receptor models, including chemical mass balance (CMB) and positive matrix factor Analysis (PMF), principal component analysis (PCA) method, and diagnostic PAHs ratio method [8–11]. However, to use receptor models and the PCA method, it is impossible to obtain emission information of vehicles and around the sampling site at that time. The diagnostic PAHs ratio method has problems, such as the inability to calculate the source contribution ratio to PAHs; the main source often differs depending on the type of PAH pair used. Moreover, this method cannot be applicable to combustion-derived particulate and NPAHs [12]. Recently, our group developed a method (the NP method) to quantify the combustion-derived fractions ($P_c$) and noncombustion-derived fractions ($P_n$) in PM. We further divided $P_c$ into high-temperature combustion-derived particulate ($P_h$) and low-temperature combustion-derived particulate ($P_l$). Major sources of $P_h$ and $P_l$ are vehicles and coal and biomass combustion. The main idea for this method is that the formation of NPAHs relative to PAHs will increase relative to combustion temperature, as the formation of nitrogen oxides and the subsequent nitration of PAHs depends on the combustion temperature. The NP method can be used for PM generated at any place and time because it only requires pyrene (Pyr) and 1-nitropyrene (1-NP) measurements [12,13].

Herein, atmospheric samples were collected every season to determine the total suspended particulate (TSP) in Sapporo from 1990 to 2002. During this period, several important countermeasures were enacted to deal with air pollution in Japan, such as restrictions on vehicle emissions of PM and NOx gases, progress on vehicle engine performance and fuel quality [7] and the ban on spiked tires in snowy regions [14]. In addition, winter heating mostly switched from coal heating to kerosene or electric heating in Sapporo by the beginning of the 1990s. It is important to clarify how these measures have improved urban air pollution in Japan. This study was conducted to elucidate the factors causing long-term and seasonal changes in TSP and $P_c$. TSP was divided into $P_c$ and $P_n$, and $P_c$ was further classified into $P_h$ and $P_l$ using the NP method. The study also examined the contributions of vehicles as well as coal and biomass combustion to $P_c$ by analyzing organic source markers.

## 2. Materials and Methods

### 2.1. Sampling

TSP samples were collected in Sapporo (the capital city of Hokkaido, Japan, population = 1,970,000). The average annual, August, and January temperatures were 9.2 °C, 22.3 °C, and −3.2 °C, respectively, and the month with the greatest average snowfall (97 cm) was February [15]. A high-volume air sampler was placed on the roof of the three-story building of the Research Institute of Energy, Environment and Geology, Hokkaido Research Organization, which is located in a residential area approximately 2 km northwest of downtown Sapporo. The 24-h TSP was collected once a week using a quartz fiber filter at an airflow rate of 1.0–1.5 m$^3$ min$^{-1}$ from the spring of 1990 to the autumn of 2002. Filters of 1432 days were obtained during the period. After sampling, filters were stored in a freezer (−20 °C) until analysis. Filters (of 12 to 13 TSP samples) were grouped based on the four seasons of the year, winter (December of the preceding year to February), spring

(March to May), summer (June to August), and autumn (September to November). In total, 51 seasonal TSP samples were obtained over the monitoring period.

### 2.2. Quantification of Organic Compound Markers

One-eighth of the filters used to collect the TSP samples in each season of the year were cut and placed in a 500-mL glass flask containing dichloromethane (400 mL). The samples were sonicated for 15 min. The extract was filtered twice through three sheets of cellulose fiber filters (Advantec, Tokyo, Japan, No. 5C). The filtrate was evaporated until less than 1 mL remained. Then dichloromethane was added to increase the total volume to 25 mL. An aliquot of the solution was used in the analysis of the chemical compounds described below.

Nine PAHs, fluoranthene (Flu), Pyr, benz[*a*]anthracene, chrysene, benzo[*b*]fluoranthene, benzo[*k*]fluoranthene, BaP, benzo[*ghi*]perylene, and coronene, were quantified according to USEPA methods using a high-performance liquid chromatograph (HPLC) equipped with a fluorescence detector [16]. Deuterated Pyr (Pyr-$d_{10}$) and deuterated BaP (BaP-$d_{12}$) were used as internal standards. A reversed-phase column (Inertsil ODS-P, internal diameter (ID) × length = 4.6 × 250 mm, GL Sciences Inc., Tokyo, Japan) was used as the analytical column. Compounds were separated using a mobile phase gradient of an acetonitrile–water mixture at a flow rate of 1 mL/min. The excitation and emission wavelengths of the fluorescence detector were optimized for each particular PAH.

Four NPAHs, 1-NP, 1,3-, 1,6, and 1,8-dinitropyrenes, were quantified using an HPLC equipped with a reducer column packed with Pt/Rh-coated particles and a chemiluminescence detector, using 2-fluoro-7-nitrofluorene as an internal standard. Reverse phase columns (Cosmosil 5C18-MS-II, 4.6 (ID) × (250 + 150) mm, Nacalai Tesque, Kyoto, Japan) were used as the analytical columns, and the mobile phase was a mixture of an acetonitrile–imidazole buffer. Further, an acetonitrile solution containing bis(2,4,6-trichlorophenyl)-oxalate and hydrogen peroxide was used as the chemiluminescence reagent solution at a flow rate of 1 mL/min [17–19].

17α(H)21β(H)-30-norhopane, 17α(H)21β(H)hopane, and 17α(H)-22,29,30-trisnorhopane were quantified using gas chromatography–mass spectrometry (GC–MS, GC: Agilent 7890B, MS: Agilent 5977B, Agilent Technologies Inc., Hachioji-shi, Japan). A DB-5MS capillary column was used as the analytical column (length × ID = 30 m × 0.25 mm, film thickness = 0.25 μm, Agilent Technologies Inc.). Three sugars, i.e., levoglucosan, mannosan, and galactosan, as well as pinonic acid, were quantified by GC–MS coupled with a solvent extraction–silylation method. The analytical conditions are adopted from previous studies [20–22] with small modifications. Samples used for the analysis of three hopanes were obtained from other TSP samples collected at the same site over the same period.

### 2.3. Calculation of Source Contributions Using the NP Method

In the NP method, TSP in the urban atmosphere is divided into $P_c$ and $P_n$. During the combustion process of organic matter, both the formation of nitrogen oxides in the flame gas and the subsequent formation of NPAHs from the parent PAHs are temperature dependent [23,24]. This causes an increase in the NPAH to parent PAH concentration ratios with the increase in the combustion temperature. High combustion temperatures occur in vehicle (diesel and gasoline) engines (2700 °C–3000 °C), which are significantly higher than those used in coal combustion stoves (1100 °C–1200 °C) and wood combustion stoves (500 °C–600 °C) [12]. Depending on these differences, $P_c$ was further divided into $P_h$ and $P_l$.

When the proportion of $P_h$ in $P_c$ and $P_c$ in P in the atmosphere are $x$ ($0 < x < 1$) and $y$ ($0 < y < 1$), respectively, Equations (1)–(3) can be written as follows:

$$[1\text{-NP}] = [1\text{-NP}_h][P_c]x + [1\text{-NP}_l][P_c](1 - x) \tag{1}$$

$$[\text{Pyr}] = [\text{Pyr}_h][P_c]x + [\text{Pyr}_l][P_c](1 - x) \tag{2}$$

$$y = [P_c]/([P_c] + [P_n]) \tag{3}$$

During the sampling period, petroleum and coal were the largest primary energy sources in Japan [25]. In Sapporo, most petroleum was used in transportation (viz. vehicles), whereas coal was used in power plants, factories, and household heating. We previously obtained the concentrations of 1-NP and Pyr in PM emitted from vehicles and coal boilers/stoves, which were common during the study period [12,13]. As a result, the following equations are available.

$$[1\text{-NP}_h]/[\text{Pyr}_h] = 0.425 \tag{4}$$

$$[1\text{-NP}_l]/[\text{Pyr}_l] = 0.0013 \tag{5}$$

Therefore, vehicle emissions and coal combustion were, respectively, regarded as high-temperature and low-temperature combustion sources. $x$, $y$ and $[P_c]$ can be obtained by Equations (1)–(3) with values from (4) and (5).

## 3. Results and Discussion

### 3.1. Relationship between the TSP and $P_c$ Concentrations

Atmospheric concentrations of TSP and $P_c$ calculated by the NP method were between 31–121 µg m$^{-3}$ and 2–27.2 µg m$^{-3}$, respectively, and the $P_c$/TSP ratios were 0.043–0.495 from spring 1990 to autumn 2002 (Table 1). Figure 1 shows the seasonal TSP concentrations from spring 1990 to autumn 2002 as well as the annual mean concentrations from 1991 to 2002. Figure 2 shows the $P_c$ concentrations during the same period as the TSP samples. The first-order regression equations of TSP and $P_c$ were Y = −2.600X + 5248 ($R^2$ = 0.401) and Y = −0.673X + 1353 ($R^2$ = 0.921), respectively. Both had negative slopes, suggesting that TSP and $P_c$ concentrations decreased over the study period. However, $P_n$ did not show a similar tendency to that of $P_c$. For example, the annual mean concentrations of $P_c$ and $P_n$ were 14.0 and 8.0 µg m$^{-3}$, respectively, in 1991 and 3.8 and 8.3 µg m$^{-3}$, respectively, in 2002. These results indicate that the long-term change in the TSP concentration can be mainly attributed to the change in the $P_c$ concentration.

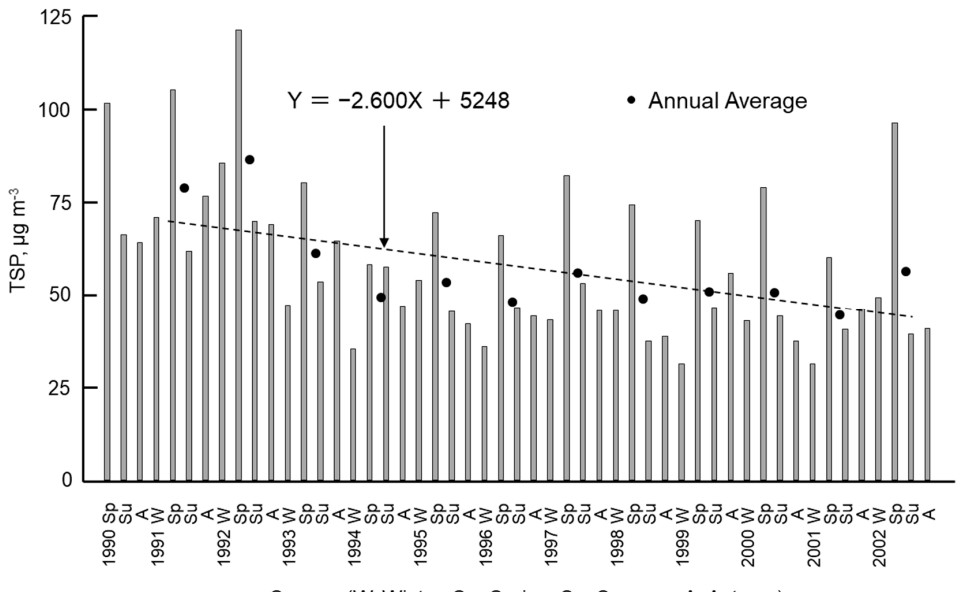

**Figure 1.** Atmospheric concentration of TSP in Sapporo from 1990 to 2002 by season.

**Table 1.** Atmospheric concentrations of TSP and $P_c$ in Sapporo (1990–2002).

| Year | Season | TSP ($\mu g\ m^{-3}$) | $P_c$ ($\mu g\ m^{-3}$) | $P_c$/TSP [a] |
|---|---|---|---|---|
| 1990 | spring | 101.6 | 5.8 | 0.057 |
| | summer | 66.0 | 3.6 | 0.055 |
| | autumn | 64.1 | 6.7 | 0.104 |
| 1991 | winter | 70.8 | 26.1 | 0.369 |
| | spring | 105.2 | 11.2 | 0.106 |
| | summer | 61.8 | 3.0 | 0.049 |
| | autumn | 76.5 | 15.6 | 0.204 |
| 1992 | winter | 85.3 | 27.7 | 0.325 |
| | spring | 121.3 | 10.0 | 0.082 |
| | summer | 69.8 | 4.0 | 0.058 |
| | autumn | 68.8 | 6.2 | 0.091 |
| 1993 | winter | 47.0 | 15.7 | 0.335 |
| | spring | 80.0 | 8.1 | 0.101 |
| | summer | 53.5 | 3.4 | 0.335 |
| | autumn | 64.4 | 10.4 | 0.161 |
| 1994 | winter | 35.2 | 17.4 | 0.495 |
| | spring | 58.2 | 8.8 | 0.151 |
| | summer | 57.5 | 3.1 | 0.054 |
| | autumn | 46.8 | 6.6 | 0.141 |
| 1995 | winter | 53.8 | 23.5 | 0.438 |
| | spring | 72.2 | 5.5 | 0.076 |
| | summer | 45.4 | 4.1 | 0.091 |
| | autumn | 42.1 | 6.3 | 0.150 |
| 1996 | winter | 35.8 | 12.5 | 0.350 |
| | spring | 65.9 | 13.7 | 0.208 |
| | summer | 46.5 | 7.7 | 0.165 |
| | autumn | 44.1 | 5.6 | 0.127 |
| 1997 | winter | 43.0 | 16.0 | 0.371 |
| | spring | 81.9 | 8.0 | 0.098 |
| | summer | 53.1 | 4.3 | 0.083 |
| | autumn | 45.5 | 7.0 | 0.154 |
| 1998 | winter | 45.6 | 15.5 | 0.340 |
| | spring | 74.3 | 5.5 | 0.074 |
| | summer | 37.3 | 7.8 | 0.208 |
| | autumn | 38.6 | 4.1 | 0.106 |
| 1999 | winter | 31.2 | 8.8 | 0.282 |
| | spring | 69.9 | 7.1 | 0.102 |
| | summer | 46.4 | 2.0 | 0.043 |
| | autumn | 55.7 | 5.9 | 0.106 |
| 2000 | winter | 42.7 | 8.5 | 0.200 |
| | spring | 78.8 | 6.1 | 0.078 |
| | summer | 44.0 | 2.3 | 0.051 |
| | autumn | 37.3 | 3.3 | 0.089 |
| 2001 | winter | 31.1 | 8.1 | 0.260 |
| | spring | 59.9 | 3.4 | 0.057 |
| | summer | 40.4 | 2.1 | 0.053 |
| | autumn | 46.0 | 2.0 | 0.044 |
| 2002 | winter | 49.3 | 5.1 | 0.104 |
| | spring | 96.2 | 5.4 | 0.057 |
| | summer | 39.2 | 2.0 | 0.050 |
| | autumn | 40.7 | 2.8 | 0.069 |

a: $P_c$/TSP = $y$ in Equation (3).

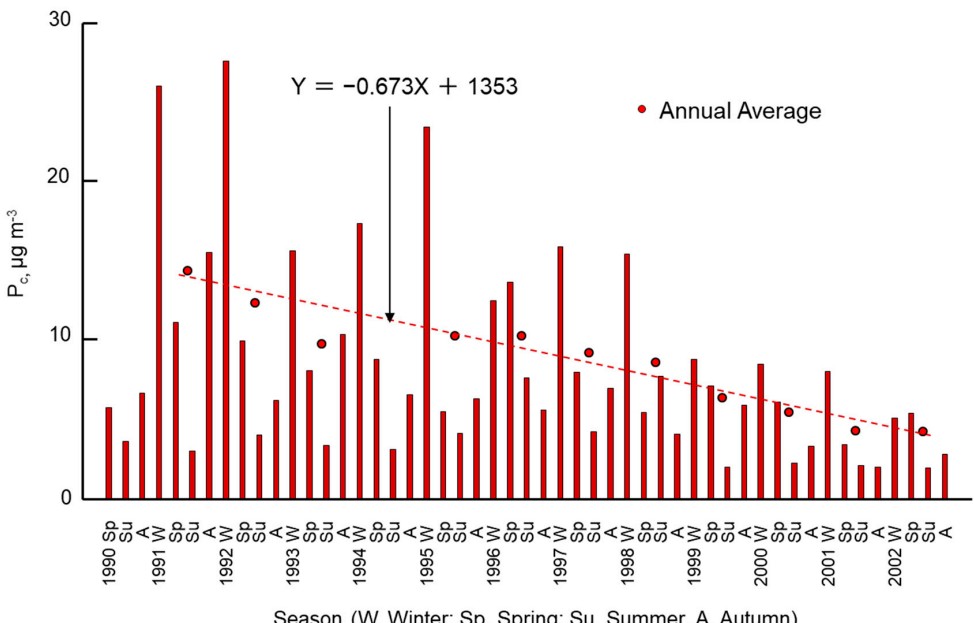

**Figure 2.** Atmospheric concentration of $P_c$ in Sapporo from 1990 to 2002 by season.

The seasonal changes in the TSP and $P_c$ concentrations differed (Table 2). The highest and lowest TSP concentrations were obtained in spring and winter, respectively, whereas the highest and lowest $P_c$ concentrations were obtained in winter and summer, respectively. However, an almost constant $P_n$ concentration was obtained in all seasons except in winter, where the concentration was 32% lower than the annual concentration. Thus, the largest and smallest fractions of $P_c$ in TSP ($P_c$/TSP) were obtained in the winter (0.324) and summer (0.075), respectively. The relative standard deviation (RSD = SD/Mean) of the annual mean $P_c$ concentration (0.732) was almost two times larger than that of TSP (0.374). These results indicate that the seasonal change in the TSP concentration was also mainly attributed to $P_c$.

**Table 2.** Seasonal atmospheric concentrations of TSP, $P_c$, and $P_n$ concentrations as well as $P_c$/TSP ratio in Sapporo.

| Season | TSP ($\mu$g m$^{-3}$) [a] | $P_c$ ($\mu$g m$^{-3}$) [a] | $P_n$ ($\mu$g m$^{-3}$) [a] | $P_c$/TSP |
|---|---|---|---|---|
| Winter | $47.6 \pm 16.2$ | $15.4 \pm 7.4$ | $5.9 \pm 3.6$ | 0.324 |
| Spring | $82.0 \pm 18.9$ | $7.6 \pm 2.8$ | $9.4 \pm 4.1$ | 0.093 |
| Summer | $50.8 \pm 10.4$ | $3.8 \pm 1.9$ | $9.4 \pm 3.1$ | 0.075 |
| Autumn | $51.6 \pm 12.9$ | $6.4 \pm 3.5$ | $10.8 \pm 3.0$ | 0.124 |
| Annual | $58.2 \pm 20.2$ | $8.2 \pm 6.0$ | $8.7 \pm 4.2$ | 0.141 |
| RSD [b] | 0.347 | 0.732 | 0.483 | |

a: Mean $\pm$ SD standard deviation (SD), b: Relative SD = (Mean/SD).

### 3.2. Factors Affecting the $P_c$ Concentration

The factors involved in the long-term and seasonal changes in the $P_c$ concentration were studied. As described in Section 2.3, $P_h$ and $P_l$, which are emitted from vehicles and coal/biomass combustion, respectively, were calculated using the NP method. Herein, the fractions of $P_h$ in $P_c$ of the 51 TSP samples were calculated based on Pyr and 1-NP concentrations. The largest and smallest seasonal mean fractions of $P_h$ in $P_c$ were obtained in the summer (0.75) and in the winter (0.56), respectively (Figure 3), suggesting that the contribution of vehicles to $P_c$ was always larger than those of coal and biomass combustion in Sapporo. This is despite the increase in heating from coal and firewood combustion in the winter and the increase in post-harvest biomass combustion in the autumn.

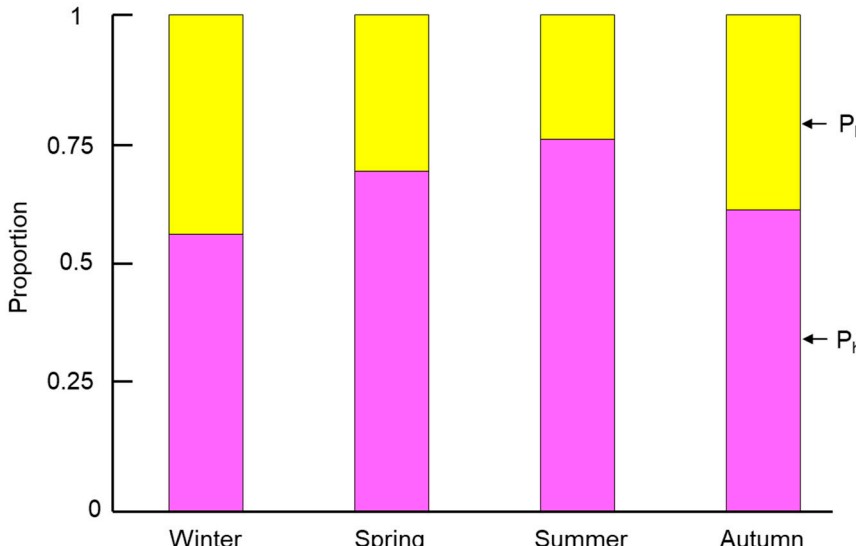

**Figure 3.** Fractions of $P_h$ and $P_l$ in atmospheric $P_c$ in Sapporo over the four seasons. Symbols: $P_h$, emitted from the high-temperature combustion source (vehicles);, $P_l$, emitted from low-temperature combustion sources (coal and biomass combustion).

In Japanese cities, the main sources of PAHs and NPAHs are vehicle emissions as well as coal and biomass combustion. Among these primary sources, vehicles emit larger amounts of NPAHs and hopanes [22,26]. Biomass combustion emits several sugars, such as levoglucosan, mannosan, and galactosan, whereas coal combustion emits much smaller amounts. Several organic acids, such as pinonic acid, also form in the atmosphere [27]. Therefore, PAHs, NPAHs, and hopanes are markers of $P_h$; levoglucosan, mannosan, galactosan, and PAHs are markers of $P_l$; and pinonic acid is a marker of $P_n$.

Figure 4 shows that the atmospheric concentrations of the total six PAHs (T-PAH), four NPAHs (T-NPAH), and three hopanes (T-Hopane) decreased over the study period, showing the same seasonal change (high in winter and low in summer) as that of $P_c$ in Sapporo. Figure 5 shows that the atmospheric concentrations of levoglucosan, mannosan, and galactosan also decreased in the same period. However, they show different decreasing rates. The relative slope, i.e., the slope of equation/average concentration, of the first-order linear regression equation was calculated for each marker compound. Relative slopes of T-PAH, T-NPAH, and T-Hopane are within a range of $-0.131$/year to $-0.099$/year, which is steeper than that of $P_c$ ($-0.082$/year). However, the relative slopes of levoglucosan, mannosan, and galactosan are in a range of $-0.036$ to $-0.040$/year, which is less steep than that of $P_c$ (Table 3). The pinonic acid concentration, whose relative slope (0.007/year in Table 3) is close to zero, did not show such a long-term change (Figure 5). Moreover, the correlation coefficients between the T-PAH, T-NPAH, and T-Hopane concentrations and the $P_c$ concentration are in the range of 0.8898–0.9433, which are significantly larger than the other chemicals (Table 4).

The total atmospheric concentrations of T-PAH, T-NPAH, and T-Hopane were significantly lower than that of $P_c$ ($\leq 1\%$ of $P_c$), and the three sugars and the pinonic acid exhibited low total concentrations similar to that of T-PAH. Despite the very small fractions of these compounds in $P_h$, their strong correlations with $P_c$ indicate that vehicles were a main contributor to the long-term decrease in the $P_c$ concentration in Sapporo. However, the contribution of coal and biomass combustion to this decrease was not as strong as that of vehicles. Since the 1990s, the Japanese government has started to gradually strengthen the PM/$NO_x$ emission regulations for new vehicles. For PM emissions from heavy-duty diesel vehicles, the regulation value in 1999 was reduced to 36% of the 1994 level. The NOx regulation value was tightened several times since 1974. It was reduced to 52% in 1989 and 33% in 1999 [28]. This countermeasure decreased the urban atmospheric concentrations of

NPAHs and PAHs by approximately 1/10 over a decade [29]. Thus, the decrease in $P_c$ in Sapporo can be mainly attributed to the PM/NOx emission regulations for new vehicles, which are reflected in the significant decrease in atmospheric concentrations of T-PAH and T-NPAH (Figure 4). Several decades ago, in the colder regions of Japan, numerous vehicles were equipped with spiked tires in winter and early spring, which significantly increased the PM concentration in the urban atmosphere in spring. The atmospheric TSP concentration was higher than 100 μg m$^{-3}$ in Sapporo in the winters from 1990–1992 and decreased quickly to 80 μg m$^{-3}$ or less over three years from 1993 to 1996 despite the high $P_c$ concentrations (Figure 1). This can be attributed to the regulation of spiked tires, which was started by the Sapporo city government in 1989 [14]. After the ordinance came into effect, the percentage of vehicles with spiked tires in Sapporo city decreased from 48.6% in 1990 to 2.4% in 1993.

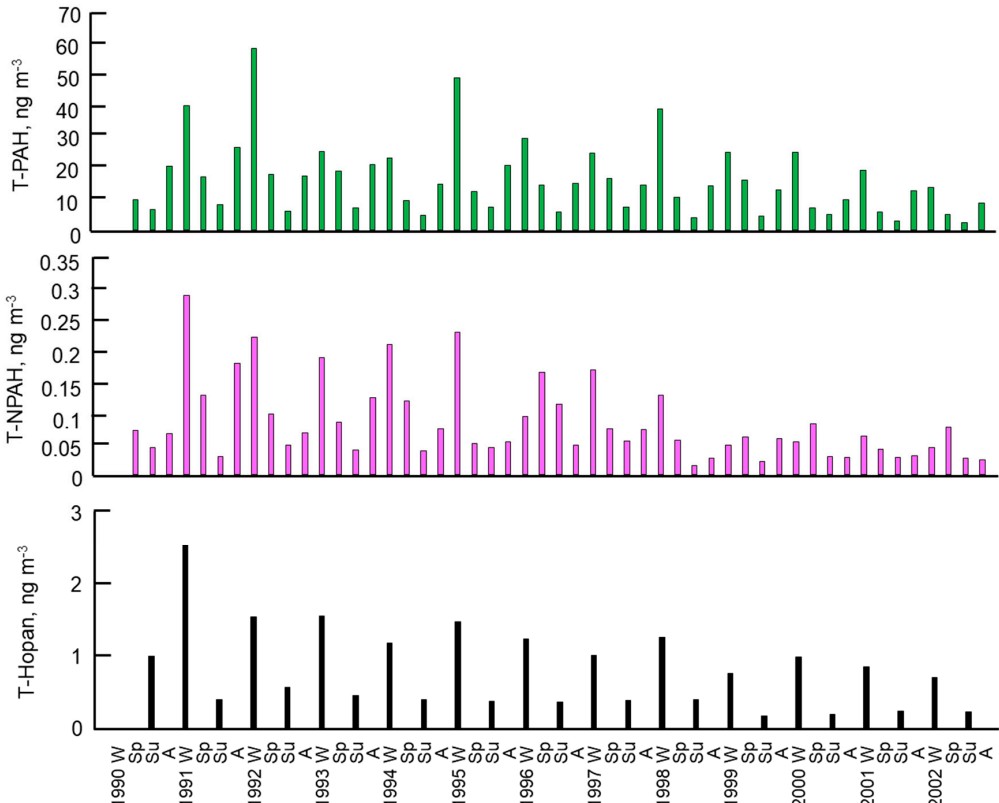

**Figure 4.** Atmospheric T-PAH, T-NPAH, and T-Hopane concentrations in Sapporo from 1990 to 2002 by season. Abbreviation: T-PAH = fluoranthene + Pyr + benz[*a*]anthracene + chrysene + benzo[*b*]fluoranthene + benzo[*k*]fluoranthene + benzo[*a*]pyrene + benzo[*ghi*]perylene + coronene, T-NPAH = 1-NP + 1,3-dinitropyre + 1,6-dinitropyre + 1,8-dinitropyre, T-hopane = 17α(H)21β(H)-30-norhopane + 17α(H)21β(H)hopane + 17α(H)-22,29,30-trisnorhopane.

All the combustion source markers exhibited the same seasonal changes as that of $P_c$, i.e., the highest and lowest concentrations are observed in winter and summer, respectively (Figures 4 and 5). There are several factors for this. First, the largest and smallest fractions of $P_l$ in $P_c$ were observed in the winter (44%) and the summer (25%), respectively (Figure 3). Moreover, the highest $P_c$ concentration is observed in the winter (Table 2). These results indicate an increase in the $P_l$ emitted from fuel combustion for winter heating. The combustion temperatures of firewood and kerosene used as fuels for heating are equal to or lower than that of coal heating (1100 °C–1200 °C). This suggests that firewood and kerosene combustion for heating emits $P_l$ that contains significantly smaller (NPAH)/(PAH) ratios than those from coal combustion for heating [12]. Second, the fuel consumption in vehicles worsens in winter. This increases the $P_h$ emissions. Furthermore, PM tends to stay on the

ground surface in the winter due to the formation of an atmospheric inversion layer [26]. These anthropogenic and meteorological factors caused long-term and seasonal changes in $P_h$ and $P_l$, which might be then reflected in $P_c$.

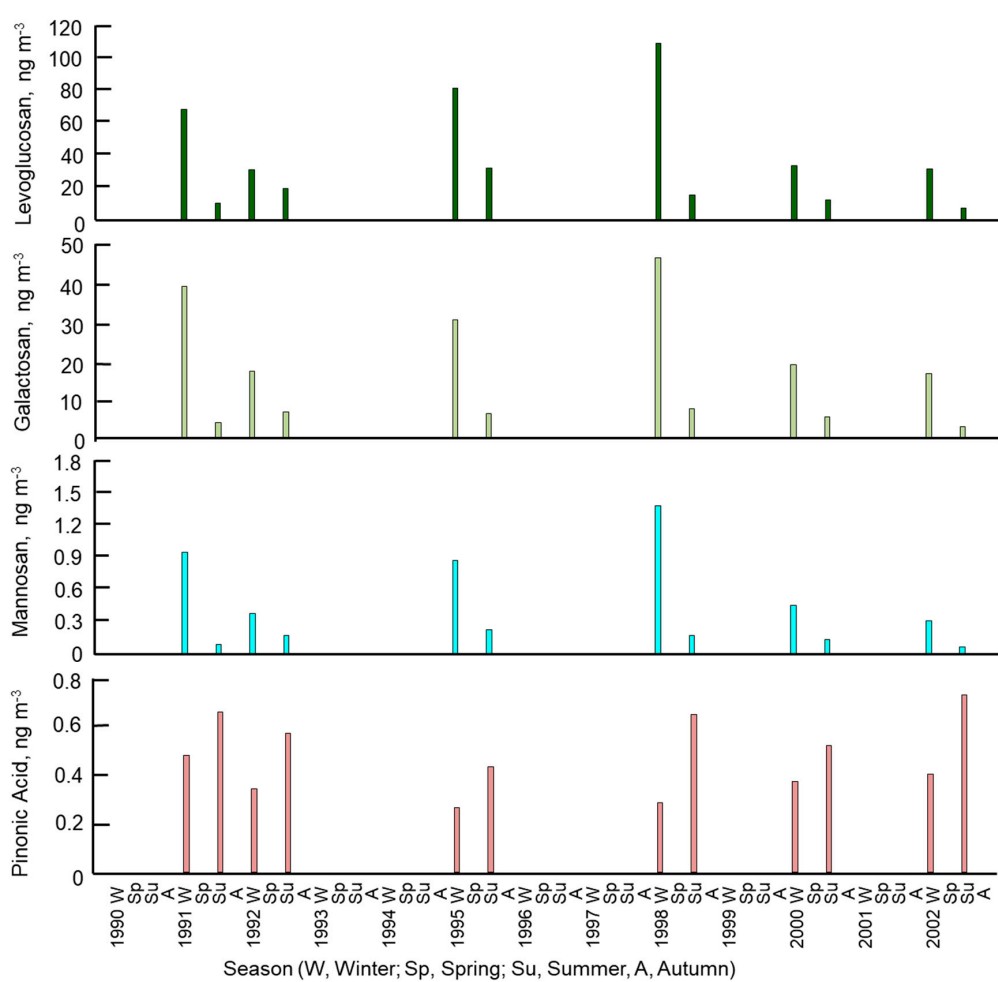

**Figure 5.** Atmospheric levoglucosan, galactosan, mannosan, and pinonic acid concentrations in Sapporo from 1990 to 2002 by season.

**Table 3.** Parameters of the first-order linear regression equation for the source markers.

| Organic Chemical | Major Source | Equation (Unit of X) | Relative Slope [a] |
|---|---|---|---|
| TSP | | $Y = -2.600X + 5248$ ($\mu g\ m^{-3}$) | $-0.045$/year |
| $P_c$ | | $Y = -0.673X + 1353$ ($\mu g\ m^{-3}$) | $-0.082$/year |
| T-PAH | Vehicle/Coal combustion | $Y = -2.17X + 2194$ (ng m$^{-3}$) | $-0.131$/year |
| T-NPAH | Vehicle | $Y = -8.52X + 16545$ (pg m$^{-3}$) | $-0.099$/year |
| T-Hopane | Vehicle | $Y = -0.0889X + 178.3$ (ng m$^{-3}$) | $-0.123$/year |
| Levoglucosan | Biomass combustion | $Y = -1.33X + 2713$ (ng m$^{-3}$) | $-0.036$/year |
| Galactosan | Biomass combustion | $Y = -0.0650X + 131.5$ (ng m$^{-3}$) | $-0.039$/year |
| Mannosan | Biomass combustion | $Y = -0.191X + 358.8$ (ng m$^{-3}$) | $-0.040$/year |
| Pinonic acid | Secondary formation | $Y = 0.0032X - 5.961$ (ng m$^{-3}$) | $0.007$/year |

a, Relative slope = slope of the equation/average concentration.

**Table 4.** Correlation coefficients (R) of the organic chemicals with $P_c$.

| Organic Chemical | R with $P_c$ |
|---|---|
| T-PAH | 0.8937 |
| T-NPAH | 0.9433 |
| T-Hopane | 0.8898 |
| Levoglucosan | 0.6477 |
| Galactosan | 0.7271 |
| Mannosan | 0.2667 |
| Pinonic acid | −0.7648 |

This study has clarified that the suppression of PM and NOx emissions from vehicles drastically reduced the atmospheric concentrations of Pyr, 1-NP, and hopanes emitted from vehicles in Sapporo in the 1990s. However, the emissions from sources with low combustion temperatures have not decreased as quickly. In the future, technological development will be essential for them [30].

## 4. Conclusions

Atmospheric TSP samples were collected in Sapporo, Japan, every season from 1990 to 2002. $P_c$ in TSP and $P_h$ in $P_c$ was determined by the NP method, and the sources of long-term and seasonal changes in $P_h$ and $P_c$ were elucidated by analyzing organic source markers.

1. The atmospheric TSP and $P_c$ concentrations ranged from 31 to 121 $\mu g\ m^{-3}$ (Mean $\pm$ SD = 58.2 $\pm$ 20.2 $\mu g\ m^{-3}$) and from 31 to 121 $\mu g\ m^{-3}$ (Mean $\pm$ SD = 8.2 $\pm$ 6.0 $\mu g\ m^{-3}$), respectively. The rate of decrease for the latter was steeper than that of the former.
2. The $P_c$ concentration exhibited a seasonal change (highest in the winter and lowest in the summer) and was different from that of TSP (highest in spring and lowest in winter). The largest and smallest $P_c$/TSP concentration ratios were observed in winter (0.324) and summer (0.075), respectively.
3. The seasonal fraction of $P_h$ in $P_c$ was in a range between 0.56 (winter)–0.75 (summer), suggesting that the contribution of vehicles to $P_c$ was always larger than that of coal and biomass combustion.
4. The atmospheric concentrations of PAHs, NPAHs, and hopanes, which are markers of vehicle emissions, exhibited long-term and seasonal changes similar to $P_c$ with large correlation coefficients (0.9433–0.8898). However, the atmospheric concentrations of levoglucosan, mannosan, and galactosan, which are markers of emissions from coal and biomass combustion, exhibited weaker correlation coefficients with $P_c$ (0.7271–0.2667). Further, the atmospheric concentrations of pinonic acid, which is a marker of the secondary pollutant formation, did not show a similar change to $P_c$. These results suggest that the change in the $P_c$ concentration was mainly caused by vehicles rather than by coal and biomass combustion and secondary pollutant formation.
5. The significant decrease in the $P_c$ concentration over the study period is mainly attributed to the Japanese PM/NOx regulations against vehicle exhaust gases.

**Author Contributions:** Project planning and supervision: K.H.; sampling: S.S. and T.A.; analyses of PAHs and NPAHs: S.S.; analyses of other organic compounds: K.H. and T.A.; writing—review: K.H.; English editing: Scholars Editing. All authors have read and agreed to the published version of the manuscript.

**Funding:** This research was financially supported in part by a Grant in Aid for Scientific Research (No. 19H03882-03 and No. 22K19686-03) from the Japan Society for the Promotion of Science.

**Institutional Review Board Statement:** Not applicable.

**Informed Consent Statement:** Not applicable.

**Data Availability Statement:** The data presented in this study are available in this article.

**Acknowledgments:** We would like to thank F. Ikemori for his supplying standard chemicals for G C–MS analyses with effective suggestions. We express our gratitude to all researchers belonging to the Environmental and Geological Research Department, Hokkaido Research Organization and Kanazawa University, who supported air sampling and chemical analysis. We would like to thank MARUZEN-YUSHODO Co., Ltd. (https://kw.maruzen.co.jp/kousei-honyaku/) for the first English language editing and G. E. Nagato, professor of Shimane University, Japan, for the final English language check.

**Conflicts of Interest:** The authors declare no conflict of interest.

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
