# Peer review of "Sources Causing Long-Term and Seasonal Changes in Combustion-Derived Particulate Matter in the Urban Air of Sapporo, Japan, from 1990 to 2002"

_atmosphere, doi:10.3390/atmos14040646_

Round 1

Reviewer 1 Report (Previous Reviewer 1)

I am very satisfied with the updated and revised version of the manuscript. The authors have greatly improved the quality of the data presented and have addressed all my comments and concerns. I therefore recommend the publication of this research study in its present form.

Author Response

Dear Reviewer #2

     Thank you very much for your warm comment. According to your suggestion, I have revised my manuscript. Major points are described below.

  1. The English of the entire paper has been reviewed with the help of G. E. Nagato, a professor at Shimane University.
  2. The title has been added with “from 1990 to 2002” to clarify the focus of the study.
  3. Several expressions have been modified to make them easier to understand.
  4. I have corrected the mistake in the reference number pointed out by J. Yuan, MDPI Office.

The above points are shown in blue in the manuscript. I hope this will be accepted for special issue: "Feature Papers in Air Quality" of Atmosphere.

Sincerely Yours,

Kazuichi Hayakawa, Ph.D., Professor Emeritus

Institute of Nature and Environmental Technology, Kanazawa University

Reviewer 2 Report (Previous Reviewer 2)

After the author's modification, the logical structure of the article is more reasonable, the context description is more consistent, and the data and results are more accurate. However, the language expression of the article is not concise and clear, there are many grammatical and expression errors, the sentence is difficult to understand, and it is difficult to read. It is recommended that the author find an expert for polishing and correction.

Author Response

Dear Reviewer #2

  Thank you very much for your warm comment. According to your suggestion, I have revised my manuscript. Major points are described below.

  1. The English of the entire paper has been reviewed with the help of G. E. Nagato, a professor at Shimane University.
  2. The title has been added with “from 1990 to 2002” to clarify the focus of the study.
  3. Several expressions have been modified to make them easier to understand.
  4. I have corrected the mistake in the reference number pointed out by Mr. J. Yuan, MDPI Office.

The above points are shown in blue in the manuscript. I hope this will be accepted for special issue: "Feature Papers in Air Quality" of Atmosphere.

Sincerely Yours,

Kazuichi Hayakawa, Ph.D., Professor Emeritus

Round 2

Reviewer 2 Report (Previous Reviewer 2)

After the author's polishing, the language expression is more accurate and smooth, and it is easier for readers to understand. At the same time, the authors corrected the references, added a description of the numbers to the article, which made the article look more compact and reasonable.

Personally, I feel that this article has met the publication requirements of the journal, and it is recommended to accept it for further publication.

This manuscript is a resubmission of an earlier submission. The following is a list of the peer review reports and author responses from that submission.

Round 1

Reviewer 1 Report

Attached please find the comments.

Reviewer 2 Report

Distinguishing particulate matter types by specific tracers is an effective way. The NP method developed by the author's group  has certain practicality in distinguish the combustion components and non-combustion particulate matters from TSP.  Meanwhile the authors improved the NP algorithm and obtained high-temperature combustion components and low-temperature combustion components, the method is somewhat innovative. However, this article has certain deficiencies in terms of language expression, logical structure, chart display, and observation data, and I personally feel that it cannot meet the publish requirements of journal, and it is recommended that authors make revisions and resubmit the manuscript.

The main questions of this manuscript include:

1) Lines 79-85, which are the writing requirements of the article, are recommended to be removed.

2) 2.1samples section: The author states that "24-h TSP was collected once a week", and then writes that there are only 51 TSP samples, how many TSP sampling data do the authors have? When was these data collected? How do the authors obtain the seasonal sampling results?

3) The "to" and "-" in line 96 should be unified.

4) The data in Table 1 are from the begging of 1990 spring, while in line 168, the author writes that the observation began from the summer of 1990, there are contradictions, and it is recommended that the authors verify and correct the expressions. At the same time, in Table 1, TSP were high in all seasons between 1990 and 1992, while after 1993 there was a clear decline in a multiplier relationship, what is the reason?

5) It is difficult to see the linear downward trend from the data in Figure 1 and Figure 2, moreover, the expression of these two pictures is very confusing, there are both year and season changes in the figures, and it is difficult to detect the trend of change. It is recommended that authors find year-to-year changes from the average annual variation. If the authors want to express seasonal trends, it is recommended that the authors perform polynomial fitting, which may work better.

6) The data in Figure 4 start from the winter of 1990, while there is no winter observation data in Table 1, where did the data for the winter of 1990 come from?

7) In Figure 5, whether the missing seasonal data are not measured or whether there is no such tracer in the observational data?

In addition, the language of this article is inaccurate, it is difficult for readers to understand the author's intention, it is recommended that the author find an English professional to polish, and make changes to the entire article before re-submitting.